# Bimodal Learning Engagement Recognition from Videos in the Classroom

**DOI:** 10.3390/s22165932

**Published:** 2022-08-09

**Authors:** Meijia Hu, Yantao Wei, Mengsiying Li, Huang Yao, Wei Deng, Mingwen Tong, Qingtang Liu

**Affiliations:** 1Hubei Research Center for Educational Informationization, Faculty of Artificial Intelligence in Education, Central China Normal University, Wuhan 430074, China; 2Huanggang High School of Hubei Province, Huanggang 438000, China; 3School of Management, Wuhan College, Wuhan 430212, China

**Keywords:** learning engagement, classroom videos, deep learning, bimodal

## Abstract

Engagement plays an essential role in the learning process. Recognition of learning engagement in the classroom helps us understand the student’s learning state and optimize the teaching and study processes. Traditional recognition methods such as self-report and teacher observation are time-consuming and obtrusive to satisfy the needs of large-scale classrooms. With the development of big data analysis and artificial intelligence, applying intelligent methods such as deep learning to recognize learning engagement has become the research hotspot in education. In this paper, based on non-invasive classroom videos, first, a multi-cues classroom learning engagement database was constructed. Then, we introduced the power IoU loss function to You Only Look Once version 5 (YOLOv5) to detect the students and obtained a precision of 95.4%. Finally, we designed a bimodal learning engagement recognition method based on ResNet50 and CoAtNet. Our proposed bimodal learning engagement method obtained an accuracy of 93.94% using the KNN classifier. The experimental results confirmed that the proposed method outperforms most state-of-the-art techniques.

## 1. Introduction

Learning engagement is vital for student satisfaction and assessment of learning effectiveness [1]. Studies on student engagement, such as definition, characterization model, and recognition method, have been research hotspots in education. Initially, learning engagement was externalized to represent learners’ positive, concentrative, and persistence state in learning [2]. Over the years, researchers have begun to accept that multidimensional components define learning engagement; now, the three-dimensional representation model, including the behavior, cognition, and emotion dimensions, proposed by Fredricks et al. [3] has been one of the most accepted and frequently used in education research. Among the three dimensions, behavioral engagement focused on students’ actions to access the curriculum, such as displaying attention and concentration or asking questions. Cognitive engagement refers to internal processes, whereas only the emotional and behavioral components are manifested in visible cues. Emotional engagement includes affective reactions, such as boredom, curiosity, and so on. Students’ engagement in the classroom is essential, as it improves the overall class learning quality and academic progress. However, measuring learning engagement is challenging in a synchronous learning environment such as the classroom. Currently, the popular learning engagement measuring methods can be divided into two categories: manual and automatic methods.

Manual methods: In a traditional classroom, learning engagement can mainly be obtained by manual measurements, such as self-reports, interviews, and observational checklists. Self-reports are questionnaires in which students describe their attention, distraction, or excitement level after the lesson [4,5]. Self-reports are practical and cheap but easily lead to biases in retrospective recall [6]. Interviews refer to obtaining students’ psychological, emotional, and behavioral characteristics through the discussion between teacher and students. Observational checklists are completed by external observers, such as teachers, to evaluate students’ performance based on a series of relevant questions regarding the factors of engagement. Interview and observational checklists are helpful but require a lot of time and effort from students and observers.

Automatic methods: Early studies aimed at estimating learning engagement based on log files and sensor data in affective computing. Measuring learning engagement on log files, such as students’ reaction times, errors, and performance [7,8,9], has been dubbed “engagement tracing” [10,11]. Measuring learning engagement on sensor data is based on physiological and neurological sensor data (i.e., blood pressure, EEG, heart rate, and galvanic skin response). Currently, with the rapid development of computer vision, deep learning methods [12,13], such as convolutional neural network (CNN), have received more attention due to their impressive recognition results on public datasets [14,15]. The learning engagement recognition method based on computer vision can extract students’ nonverbal cues (i.e., emotion [16], head gaze [17], and gesture [18]) from classroom videos to automatically recognize learning engagement in the real classroom. The nonverbal visual cues from videos extracted by the high-definition camera installed in the classroom [19] can provide students’ behavioral, physiological, and psychological information and have time continuity, which is helpful in exploring the internal regularity of learning engagement. The automatic methods are non-invasive, effective ways to automatically monitor learning engagement in many learning environments, especially real classrooms. It can help teachers improve their instructional strategies and maintain the right level of interactions so that students can easily adapt to teachers’ styles for better understanding and learning.

Even though automatic methods based on computer vision obtain impressive performance, they often need datasets when training deep learning algorithms. There are few public engagement datasets in a real classroom, and their modalities of engagement are single. Furthermore, the complexity and occlusion in a real classroom make it difficult to detect each student, even with high-definition cameras installed. Exploring learning engagement with only a single mode is challenging in this situation. Currently, recognizing and analyzing the students’ engagement based on multiple nonverbal cues has been the trend. For example, Ventura et al. [20] analyzed students’ faces, body postures, and the classroom environment and obtained the engagement of a single student and the whole classroom. Ashwin et al. [21] proposed a CNN architecture for unobtrusive engagement analysis using non-verbal cues. Süme et al. [6] explored classifying engagement by training Attention-Net for head pose estimation and Affect-Net for facial expression recognition using facial videos. In a real classroom, the unobtrusive learning engagement can be effectively recognized using multiple nonverbal cues, such as facial expressions, hand gestures, and body postures. The contributions of this paper are shown below:

Constructing a multi-cues classroom engagement database. Based on non-invasive classroom videos, we constructed a learning engagement dataset. Our self-built dataset contains multiple nonverbal cues, including emotional and behavioral cues, such as the students’ facial expressions, hand gestures, and body posture.Using the power IoU loss based on YOLOv5 to detect the students in the classroom and considering the complexity of an actual classroom, we introduced the  α-IoU loss function in YOLOv5 to detect the students and obtained effective detection results.Proposing a bimodal engagement recognition method based on a self-built dataset, we applied ResNet50 for recognizing student emotional engagement. We used the self-built behavioral engagement dataset to train the CoAtNet network to estimate student behavioral engagement.

This paper is organized as follows. Section 2 introduces the related works of this paper. Section 3 describes the proposed methodology of the bimodal engagement recognition method. Section 4 shows the detailed experiment and results. Section 5 presents the conclusion and future work.

## 2. Related Works

### 2.1. YOLOv5

As for object detection, CNN is the leading technique that has obtained high recognition accuracies. The object detection methods based on CNN can be mainly divided into two categories: two-stage detectors and one-stage detectors. Among the detectors, two-stage detectors usually have higher accuracies, while one-stage detectors perform better in computing speed. The complexity of the offline classroom environment puts forward a higher requirement for real-time student detection. One-stage detectors are favorable algorithms in this case due to the more simplified network design and faster inference speed. As one of the most popular one-stage detectors, YOLO series algorithms have been widely used in deep learning [22,23,24]. Over the years, after many times of algorithm optimization and improvement, YOLO series algorithms have obtained the best accuracy and speed performance [25].

YOLO families are often used for object detection tasks and have achieved impressive detection results [26]. The comparison of YOLO algorithms in different versions is shown in Table 1. YOLO obtained a fast detection speed segment by discarding the step of generating suggestion boxes, thus reducing calculation and time consumption. Since only the last layer of feature maps was used, the model had a poor detection performance on small objects. To improve the detection accuracy, YOLOv2 and YOLOv3 introduced an anchor frame mechanism, in which shallow feature maps and high-level feature maps were used to detect small and large objects, respectively. YOLOv2 uses Darknet19, which borrows a priori frame from the RPN network, to maintain the detection speed while improving the model’s accuracy. YOLOv3 uses a 53-layer convolutional network for feature extraction to obtain three different sizes of feature maps. YOLOv4 uses CSPDarknet53 as the backbone network and CIOU_LOSS for prediction boxes screening, which improves the detection accuracy. YOLOv5 also uses CSPDarknet53, but the neck network has adopted the feature pyramid network (FPN) and pixel aggregation network (PAN) structures. With a lightweight model size, it is comparable to YOLOv4, in terms of accuracy, but superior to YOLOv4 in speed. As the latest version of the YOLO algorithm, YOLOv5 has proven to improve the processing time of deeper networks significantly. As Figure 1 shows, YOLOv5 comprises three main parts: the backbone for extracting features, the neck for fusing elements, and the head for detecting the object.

Backbone: As for the core of the YOLOv5 model, the backbone extracts feature from the input image for subsequent utilization and processing. Specifically, backbone includes the focus, CSP (cross-stage partial) [27], and SPP (spatial pyramid pooling) modules. The focus module slices one data information into four and then generates them on the channel dimension to reduce the model’s flops and increase the training speed. There are two CSP modules in the YOLOv5 model: CSP1_X is applied in the backbone part, while CSP2_X is in the neck. The CSP modules are designed to reduce the amount of computation while maintaining precision by reducing model size. The SPP module improves the receptive field of the network by converting any size of the feature map into a fixed-size feature vector.

Neck: The neck is designed to enhance the ability of feature fusion by adopting FPN (feature pyramid networks) and PAN (path aggregation network). The FPN structure conveys strong semantic features from the top feature maps into the lower feature maps. To enhance the method’s robustness, the model extracts features by employing a PAN structure that strengthens the bottom-up path and improves the propagation of low-level information.

Head: This part generates anchor boxes for feature maps and outputs final output vectors with class probabilities and bounding boxes of detected students.

YOLOv5 was pre-trained on the COCO dataset, a public image dataset for object recognition, segmentation, and labeling. This dataset contains more than 200,000 labeled images, which belong to about 80 different classes and include the *person* class. Therefore, YOLOv5 can be used as such to detect students in the classroom. The YOLOv5 network released five different models, i.e., YOLOv5n (smallest), YOLOv5s, YOLOv5m, YOLOv5l, and YOLOv5x (largest). This paper uses the YOLOv5n lightweight models for student detection in the classroom.

### 2.2. ResNet50

The proposal of ResNet makes it possible to obtain good performance and efficiency of the network, even when the network develops in a deeper direction [28]. The main component of ResNet is the residual module. The residual networks are deep neural networks that follow the basic idea of skipping blocks using shortcut connections. It is one of the cores in the classic computer vision task, which is widely used in object classification. The presentation of the ResNet network well-solves the degradation and overfitting problems caused by the increasing number of layers in the network.

The deep residual learning framework of ResNet is shown in Figure 2. The residual module consists of two dense layers and a skip connection. The activation function of each of the two dense layers is the ReLU function. The main idea of ResNet is to add a direct connection channel in the network, called a highway network, which allows the original input information to be passed directly into the next layer. The classic ResNet model includes ResNet50, ResNet18, ResNet101, and so on. It has been indicated that ResNet50 performs better in image scene classification than other CNN models in the ImageNet datasets [29]. ResNet50 is composed of 50 layers deep and over 25.6 million parameters. The set combines convolution, identity block (input = output), and a fully connected layer. The identity x corresponds to the input value of the original block or signal. The output value of the residual block is the sum of the input value of the block and the output values of the internal layers of the block.

### 2.3. CoAtNet

Since AlexNet obtained excellent performance in the ImageNet classification task [30], the CNN model has been the leading architecture in computer vision. Meanwhile, self-attention models, such as transformers [31], have attracted increasing interest and shown a higher model capacity at scale than CNN. However, the generalization of transformers might be worse than traditional CNN when transformers have a larger model capacity. Proposed by Dai et al. [32] in 2021, CoAtNet is an effective model that inherits the great generalization property of ConvNets and enjoys the superior scalability of the transformer models by combining the strengths of depthwise convolution and self-attention. The architecture of the CoAtNet network is shown in Figure 3. To optimally balance the model generalization and capacity, the CoAtNet network first merges the convolution and self-attention within one basic computational block. Then, it vertically stacks different types of computational blocks together to form a complete network.

Merging Convolution and Self-Attention: The merging core of convolution and self-attention is that both the FFN module in the transformer and MBConv employ the design of “inverted bottleneck”. In addition, depthwise convolution and self-attention can be expressed as a per-dimension weighted sum of values in a pre-defined receptive field; the receptive field size is one of the most crucial differences between self-attention and convolution. Based on the above, CoAtNet can simply sum a global static convolution kernel with the adaptive attention matrix, either after or before the Softmax normalization.

Vertical Layout Design: CoAtNet vertically stacks ConvNets modules and attention modules with a multi-stage layout of C-C-T-T, where C and T denote convolution and transformer, respectively. CoAtNet employed the MBConv block as the main component rather than the residual block, and the transformer blocks were placed in the last two stages rather than the last stage. The C-C-T-T layout is designed to reduce the spatial size and improve the generalization, capacity, and efficiency by using down-sampling.

## 3. Proposed Method

In this paper, based on non-invasive classroom videos, first, we created a multi-cues classroom learning engagement database. Next, we introduced the power IoU loss function-based YOLOv5 to detect the students. Then, we designed a bimodal learning engagement recognition method based on ResNet50 and CoAtNet. Finally, this paper builds a classifier to estimate three-level learning engagement automatically.

### 3.1. Data Collection and Annotation

#### 3.1.1. Data Collection

The dataset is a collection of videos from 28 students (6 male and 22 female) during regular lessons at a university. Before the experiment, we obtained consent from the teacher and students that their performance in the smart classroom would be videotaped. The equipment of the smart classroom is shown in Figure 4. The smart classroom is equipped with several cameras positioned above the teacher’s head around the blackboard area of the classroom. In the back of the smart classroom, there is a one-way mirror, behind which is an observation room equipped with the necessary hardware to receive and record audio and video data. In the observation room, researchers and educators can observe the teacher or students non-intrusively.

We obtained 12 videos of 45 min duration in MP4 format. To get the video frames, we extracted images from the videos 1 frame with 6 s. The total number of samples generated from 12 videos is around 4550.

#### 3.1.2. Data Annotation

In this study, we categorized learning engagement levels as low, medium, or high. The engagement state details are mentioned below. Our engagement states include emotional and behavioral aspects, such as the students’ facial expressions, hand gestures, and body posture.

Low engagement (EL 1): Student is not thinking about the learning task, eyes barely opening, yawning, looking away, body bending on the desk, head fully lying on hands or desk, with negative emotions—e.g., boredom, sleepy.Medium engagement (EL 2): Student is thinking about the learning task, body leaning forward, head supported by a hand, with no expression on the face.High engagement (EL 3): Student is engaged in the learning task, body leaning forward, looking at the teacher/board/book, taking notes, listening, with positive emotions—e.g., confusion and happy.

We chose the open-source software labelImg (the annotation interface of the labelImg tool is shown in Figure 5) to label the engagement level (EL) and bounding boxes for the exact location of each student. Fifteen graduate students performed the annotations. We used one bounding box for the face and one for both body postures to achieve the optimal bounding box computations. The annotated image with the class label and object localization is stored in the XML file. Each image will have three engagement level labels (emotional engagement, behavioral engagement, and overall engagement) and two sets of corresponding bounding box coordinates (one set corresponds to the face, and another corresponds to the body posture). In the study, we proposed to apply the Dawid-Skene algorithm [33] based on the expectation maximization (EM) algorithm to improve the annotation accuracy. Our annotations results correspond to the students’ self-report.

In this process, as Table 2 shows, a total of 4550 classroom images were annotated with 33,330 student images, including 12,850 low engagement labels, 12,100 medium engagement labels, and 8380 high engagement labels in the overall engagement dimension. Figure 6 shows the image samples that have been annotated with different engagement levels in the overall dimension.

### 3.2. Student Detection Based on YOLOv5 with Power Loss

#### 3.2.1. Student Detection Based on YOLOv5

Similar to the one-stage object detection algorithms, YOLOv5 runs a single convolutional network on the input image to simultaneously predict multiple bounding boxes and class probabilities for those boxes. For example, as Figure 7 shows, YOLOv5 divides the input classroom image into 7 × 7 grids. If the center of a student falls into a grid cell, the grid is responsible for detecting the student. The grid at the center of the student predicts the bounding box, confidence, and probability where the individual student belongs and returns the position coordinates and prediction confidence. The confidence scores will be zero if no object exists in that cell. At the output of the YOLOv5 network, the loss function and non-maximum suppression (NMS) algorithms were used to retain the maximum value of object prediction for each student. The output classification result with the maximum probability generates a boundary box, predicts its category, and finally, gets the detection result.

#### 3.2.2. Loss Function

Bounding box (bbox) regression is a fundamental task in computer vision. So far, the most commonly used loss functions for bbox regression are the intersection over union (IoU) loss and its variants. However, the IoU loss suffers from the gradient vanishing problem when the predicted bboxes are not overlapping with the ground truth, which tends to slow down convergence and result in inaccurate detectors. The above-mentioned has motivated the design of several improved IoU-based losses, including generalized IoU (GIoU), distance IoU (DIoU), and complete IoU (CioU).

He et al. [34] present a new family of IoU losses by applying power transformations to existing IoU-based losses. It generalizes existing IoU-based losses, including GioU, DioU, and CioU, to a new family of power IoU losses for more accurate bbox regression and object detection. By modulating the power parameter α, α-IoU offers the flexibility to achieve different levels of bbox regression accuracy when training an object detector. It showed that α-IoU can improve bbox regression accuracy by up-weighting the loss and gradient of high IoU objects in YOLOv5.

In this study, we applied α-IoU as the location loss function of YOLOv5. First, calculate the minimum area of two boxes. Then, calculate the size of the closed area that does not belong to two boxes and calculate the IoU. Finally, powering the IoU and subtracting this part from ${\mathrm{IoU}}^{\alpha }$ to get α-IoU (it is written alphaIoU in the equation), as shown as follows:(1)$$\mathrm{IoU}=\frac{A\cap B}{A\cup B}$$
(2)$$\mathrm{alphaIoU}=\frac{1-{\mathrm{IoU}}^{\alpha }}{\alpha },\;\alpha >1\;$$
where A is the ground truth, and B is the prediction box. IoU is the Jaccard overlap calculation. Our total loss function is shown as:(3)$$L_{\mathrm{total}}=L_{\mathrm{conf}}+L_{\mathrm{cla}}+L_{\mathrm{alphaIoU}}\;$$
where $L_{\mathrm{conf}}$, $L_{\mathrm{cla}}$, and $L_{\alpha -\mathrm{IoU}}$ are the confidence loss weight, classification loss weight, and localization loss weight, respectively. The calculation equation is as follows:(4)$$\begin{array}{c} L_{\mathrm{conf}}=\lambda_{\mathrm{obj}}\overset{S^{2}}{\underset{i=0}{\sum }}\overset{B}{\underset{j=0}{\sum }}I_{\mathrm{ij}}^{\mathrm{obj}}\left[-\hat{c_{i}}{\mathrm{lnc}}_{i}-\left(1-\hat{c_{i}}\right)\ln\left(1-\hat{c_{i}}\right)\right]\\ +\lambda_{\mathrm{nobj}}\overset{S^{2}}{\underset{i=0}{\sum }}\overset{B}{\underset{j=0}{\sum }}I_{\mathrm{ij}}^{\mathrm{nobj}}\left[-\hat{c_{i}}{\mathrm{lnc}}_{i}-\left(1-\hat{c_{i}}\right)\ln\left(1-\hat{c_{i}}\right)\right] \end{array}$$
(5)$$L_{\mathrm{cla}}=\overset{S^{2}}{\underset{i=0}{\sum }}\overset{B}{\underset{j=0}{\sum }}\underset{c\in \mathrm{cla}}{\sum }I_{\mathrm{ij}}^{\mathrm{obj}}\left[-\hat{p_{i}}\ln\left(p_{i}\left(c\right)\right)-\left(1-\hat{p_{i}}\left(c\right)\right)\ln\left(1-\hat{p_{i}}\left(c\right)\right)\right]$$
(6)$$L_{\mathrm{alphaIoU}}=\overset{S^{2}}{\underset{i=0}{\sum }}\overset{B}{\underset{j=0}{\sum }}\left(1-\mathrm{alphaIoU}\right)$$
where $S^{2}$ is the number of pieces of divided grids, and B is the number of anchor boxes for each grid. $I_{\mathrm{ij}}^{\mathrm{obj}}$ and $I_{\mathrm{ij}}^{\mathrm{nobj}}$, respectively, indicate whether the jth anchor box of the ith cell contains the object. $\lambda_{\mathrm{obj}}$ and $\lambda_{\mathrm{nobj}}$ is the weight coefficient of whether the grid has targets. $c_{i}$ and $\hat{c_{i}}$, respectively, represent the weight for whether the anchor contains the object. c is the predicted category. $p_{i}\left(c\right)$ and $\hat{p_{i}}\left(c\right)$ are the predicted category and ground truth category after one-hot encoding, respectively.

### 3.3. Bimodal Learning Engagement Recognition Method

Classroom videos contain non-invasive visual cues, such as facial expression, head pose, body posture, hand gestures, etc. Among these multiple cues, facial expression is often considered the main cue of learning engagement or effective state. However, a student’s face is not always available due to occlusion in the complex classroom environment. In this case, it is difficult to analyze learning engagement through only one mode, so it is a trend to explore the classroom learning engagement of multiple modes. Consequently, we proposed a bimodal learning engagement recognition method through students’ faces and upper bodies. Figure 8 shows the structure of our proposed bimodal learning engagement recognition methodology.

We used the transfer learning method in the emotional engagement channel by applying ResNet50 as the pre-trained model and fine-tuning for recognizing student emotional engagement. In the behavioral engagement channel, we used the self-built behavioral engagement dataset to train the CoAtNet network to estimate student behavioral engagement. To automatically estimate the three-level engagement from emotional and behavioral features, we selected optimal classifiers and their parameters to build a general, person-independent engagement model, which was not over-fitted to the training data.

## 4. Experimental Results

The experimental platform was Windows 10 64bit with Intel(R) Xeon(R) Silver 4112 CPU @ 2.60 GHz, NVIDIA TITAN V (12 GB storage), CUDA Version 11.3, CUDNN v8.2. The deep learning framework was Pytorch.

### 4.1. The Result of Student Detection

In this study, the YOLOv5n network was trained by stochastic gradient descent (SGD) in an end-to-end way. The batch size of the model training was set to 32. The decay rate (decay) of weight was set to 5 × 10^−4^, learning rate was set to 0.01, and α of the α-IoU loss function was set to 3. The number of training epochs was set to 100. After training, the weight file of the obtained detection model was saved, and the test set was utilized to evaluate the model’s performance.

Figure 9 shows the mAP under the experiment with different loss functions. With the increase in the number of training epochs, the model converged very quickly, until it neared the optimization, so the mAP curve increased sharply and then tended to be stable. After reaching the steady state, the corresponding mAP value of the network based on α-IoU loss function was the highest, which means the model performance based on α-IoU loss function was the best.

Table 3 shows that, compared with existing IoU loss and IoU-based losses, including GIoU, DIoU, and CIoU, YOLOv5 with α-IoU loss obtained the most effective detection results: the precision, recall, AP, mAP@.5, and mAP@.5: 0.95 values were improved 1.4%, 1.8%, 0.6%, 0.9%, and 0.4%, respectively. The overall detection accuracy of the model was high, and each index reached more than 95%, which can meet the accuracy requirements of student detection in the classroom.

The network’s final output is the location boxes of the two varieties of student targets recognized (the prediction box of student location) and the probability of belonging to a specific category. Figure 10 and Figure 11 show the detection effects based on our proposed method. Our method can accurately identify each student without missing, and the confidence range was 0.76–1.0. The confidence scores of the predicted labels were higher, so students can be detected more accurately.

### 4.2. The Result of Emotional Engagement Classification

The model training process is composed of base training and fine-tuning. We explore the results obtained using the five pre-trained networks, viz. ResNeXt, DensNet, MoblieNet, EfficientNet, and ResNet50 classify an image into three emotional engagement levels, viz., low engagement, medium engagement, and high engagement. We have compared the results obtained by them. The self-built emotional engagement dataset comprises 10,450 low engagement images, 11,800 medium engagement images, and 11,080 high engagement images to train the network to estimate student emotional engagement. The training is done using an SGD optimizer with an initial learning rate of 0.01, decay 5 × 10^−4^, batch size 32, and momentum 0.9 for 100 epochs. The testing accuracies of different networks are given in Table 4.

Testing accuracy indicates how successful the network is in correctly classifying the data it is being trained on. Table 4 shows that the testing accuracy is the highest for the ResNet50 network with an accuracy of 87.3%. The result indicates that ResNet50 can well-solve the degradation and overfitting problems caused by the increasing number of layers in the network. Hence, it performs best in emotional engagement classification than other CNN models. The ResNeXt obtains the lowest accuracy of 76.28% because of its complex network structure, which influences model generalization.

To further evaluate the performance of the different methods, we tested the accuracy of the five networks on different categories of images; the detailed testing accuracies are shown in Figure 12. It can be seen from Figure 12 that the networks obtained different testing accuracies in different categories, and the accuracies of the DensNet, Moblienet, Efficientnet, and ResNet50 networks on low engagement were lower than that of medium or high engagement. One reason may be that the number of low engagement datasets was small, thus resulting in low characteristics ability in this category.

### 4.3. The Result of Behavioral Engagement Classification

In this section, we used the self-built behavioral engagement dataset, composed of 12,330 low engagement, 10,750 medium engagement, and 10,250 high engagement images, to train the CoAtNet network to estimate student behavioral engagement. Our self-built dataset was split into 17,031 for training, 4866 for validation, and 2433 for testing. Before training, we aligned all images to a similar size of 224 × 224. We trained the CoAtNet network using softmax cross-entropy loss to predict categorical models of engagement level: low, medium, and high. The training was done using an Adam solver, with an initial learning rate of 1 × 10^−4^ for 200 epochs. We compared the different networks (VGG16, ResNet18, and CoAtNet) trained on our self-built dataset.

As can be seen from Table 5, the results using VGG16 and ResNet18 pre-trained networks were 86.31% and 84.52, respectively, and were higher than that of VGG16 and ResNet18 without pre-trained networks. The result indicates the effectiveness of transfer learning. Meanwhile, our self-built dataset trained on CoAtNet obtains the highest accuracy of 89.97%, even than VGG16 and ResNet18, which used pre-trained networks. CoAtNet had both good generalization, similar to ConvNets, and superior model capacity, similar to transformers, thus achieving compelling performances.

The confusion matrix on testing datasets of CoAtNet is shown in Figure 13. The matrix’s horizontal and vertical coordinates represent the predicted and true labels, respectively. As can be seen from the confusion matrix, the predictions of the CoAtNet network were 93%, 88%, and 89% on the low, medium, and high engagement levels, respectively, which means that CoAtNet can well-classify the different engagement levels.

### 4.4. The Result of Decision Fusion

In this paper, we compared seven classifiers, ranging from simpler models, such as decision trees, to more complex models. The classifiers that we used included decision tree (DT), random forest (RF), naive bayes (NB), k-near neighbor (KNN), logistic regression (LR), and support vector machines (SVM). As Table 6 shows, the overall classification accuracy of the KNN classifier was 90.91%, which was 0.59%, 5.75%, 14.3%, 0.81, 6.06%, 9.07%, 6.06%, and 9.07% higher than that of the DT, NB, LR, RF, SVM (linear), SVM (poly), SVM (RBF), and SVM (Sigmoid) algorithms, respectively. The results indicate that the KNN algorithm can well-classify bimodal learning engagement.

After selecting the KNN algorithm as the optimal classifier, we further adjusted the parameters of the k values of KNN. Table 7 provides the results obtained using different values of k. We have in this paper used accuracy, precision, and recall as performance matrices to evaluate the results obtained. Precision indicates the false positives obtained, while recall gives us the false negatives. When the value of k is 2, the KNN algorithm got the best results, with 93.94% accuracy, 92.86% precision, and 92.86% recall.

## 5. Discussion and Future Work

### 5.1. Discussions

The existing works monitor students’ emotional engagement (sleepy, boredom, frustration, concentration, and so on) by analyzing students’ expressions [6,19]. A few other kinds of research mainly considered behavioral engagement (raising hands, lying on the desks, etc.) from the classroom videos [23,35]. In this paper, we propose a bimodal engagement recognition method to automatically monitor the engagement level of students in the offline classroom. We applied ResNet50 for recognizing student emotional engagement and used the self-built behavioral engagement dataset to train CoAtNet network to estimate student behavioral engagement. Dataset construction is always an essential step in student engagement analysis research. There is no public multiple nonverbal cues engagement database in the offline classroom. Based on non-invasive classroom videos, we create a learning engagement dataset that consists of 12,850 low, 12,100 medium, and 8380 high engagement labels. Our self-built dataset contains multiple nonverbal cues, including emotional and behavioral engagement aspects, such as the students’ facial expressions, hand gestures, and body posture. Considering the complexity of an actual classroom, we introduced the power IoU loss function in YOLOv5 to detect the students and obtain a precision of 95.4%

Ashwin et al. [21] proposed an unobtrusive engagement recognition method using non-verbal cues that obtained 71% accuracy, and our proposed bimodal learning engagement method obtained 93.94% accuracy on the KNN classifier. Uçar et al. [36] presented a model to predict students’ engagement in the classroom from Kinect facial and head poses. However, the range of Kinect is small and cannot be used in a large-scale classroom. Our experiment proves that student engagement can be recognized unobtrusively using non-verbal cues, such as facial expressions, hand gestures, and body postures, as captured from the frames of the classroom video.

### 5.2. Future Work

Although our proposed learning engagement recognition method achieved good results, some problems should be addressed. Firstly, the bimodal learning engagement contains emotional and behavioral dimensions. As one of the three dimensions proposed by Fredricks et al., the cognitive dimension significantly affects student engagement, though it is not easy to identify, even for human observers. In the future, we can explore the impact of the other dimension on learning engagement, such as the cognitive or social dimension. Secondly, our self-built dataset only contains visual information. In the learning process, students’ verbal, text, and physiological information can also reflect students’ engagement to a certain extent. Hence, we can also expand the dataset with other modes, such as students’ speech and text, in future research.

In addition, our proposed bimodal learning engagement recognition method is based on the decision fusion of students’ behaviors and emotions, which ignores the correlation of other features of students in the classroom. We can try other fusion methods, such as features fusion, and optimize the fusion method by combining advanced technology.

Finally, the proposed method recognizes engagement through students’ visual cues from classroom videos, which ignores the correlation of other features in the classroom environment to some extent. In the future, we suggest combining the video sequence and spatial features of student distribution using the deep learning technique [37,38].

## Figures and Tables

**Figure 1 sensors-22-05932-f001:**
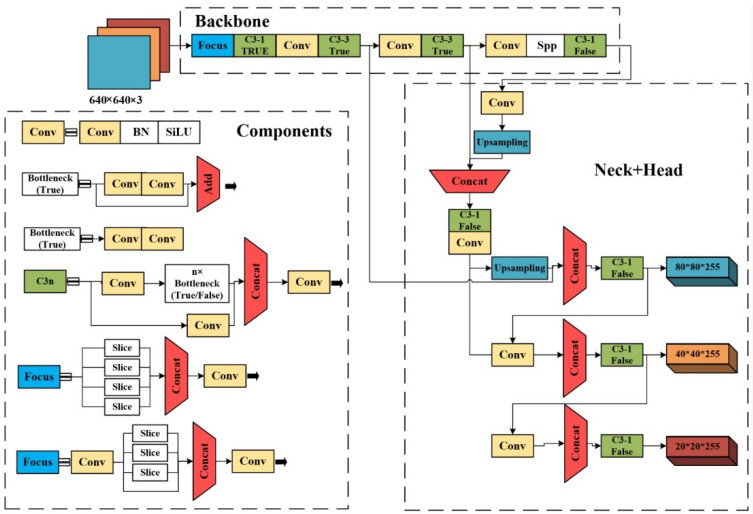
The architecture of the YOLOv5 model.

**Figure 2 sensors-22-05932-f002:**
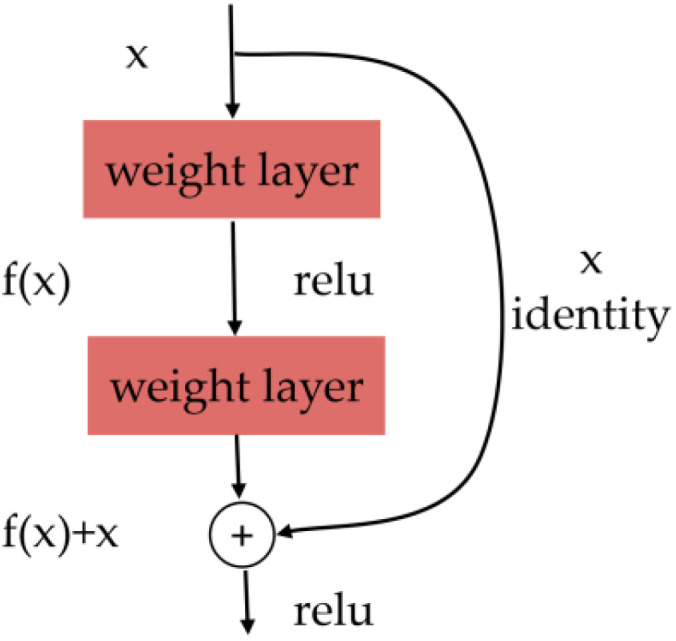
The residual module of ResNet.

**Figure 3 sensors-22-05932-f003:**
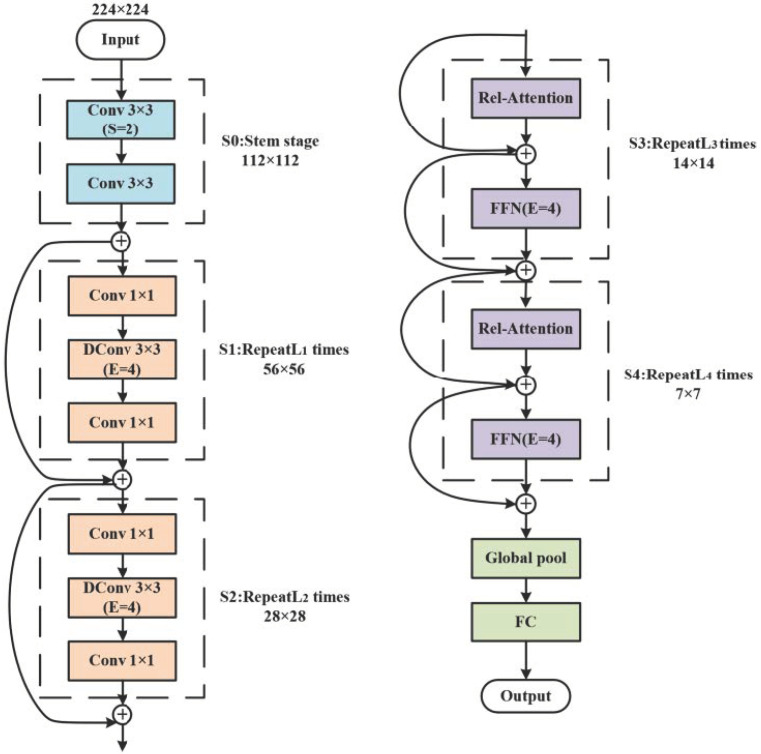
The architecture of the CoAtNet network.

**Figure 4 sensors-22-05932-f004:**
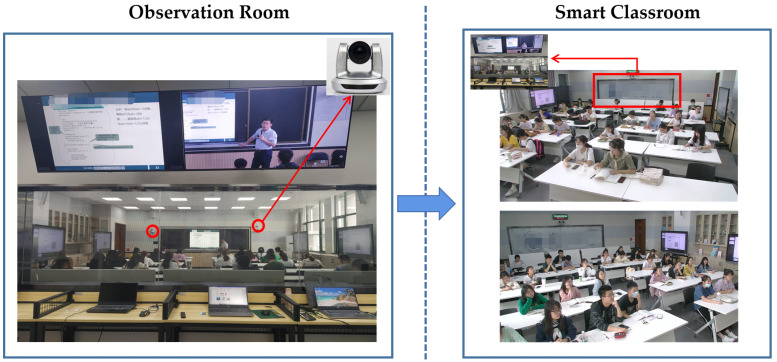
The equipment of the smart classroom.

**Figure 5 sensors-22-05932-f005:**
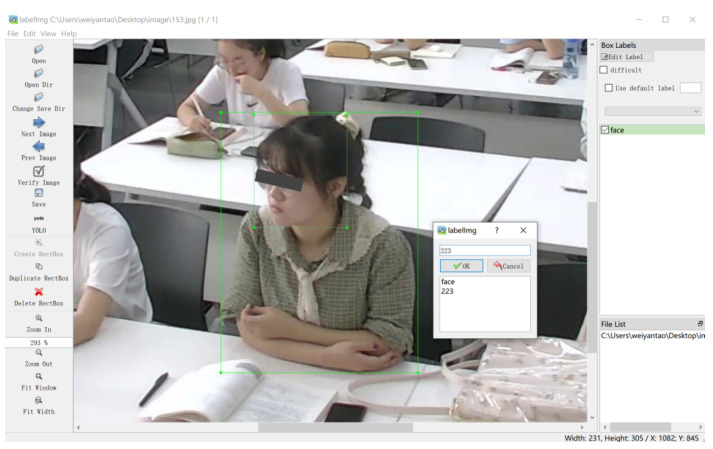
The annotation interface of the labelImg tool.

**Figure 6 sensors-22-05932-f006:**
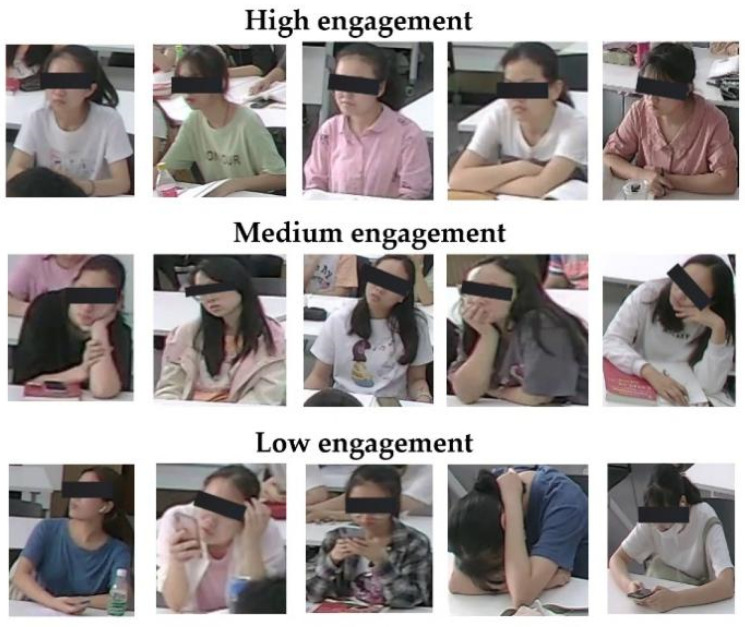
Image samples that have been annotated with different engagement levels in the overall dimension.

**Figure 7 sensors-22-05932-f007:**
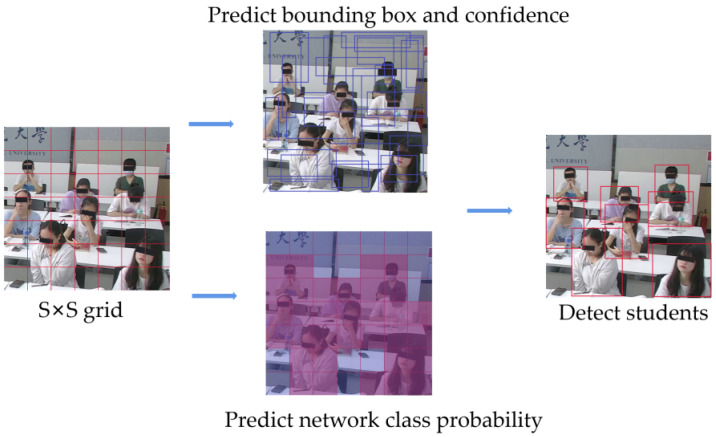
Student detection, based on YOLOv5.

**Figure 8 sensors-22-05932-f008:**
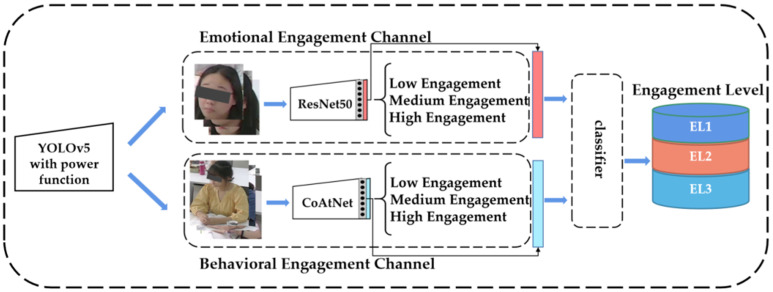
The structure of our proposed bimodal learning engagement recognition methodology.

**Figure 9 sensors-22-05932-f009:**
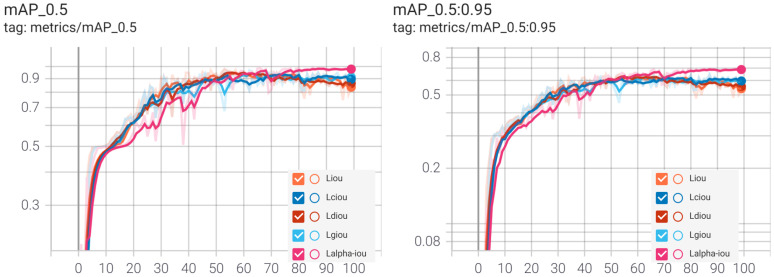
The mAP under the experiment with different loss functions.

**Figure 10 sensors-22-05932-f010:**
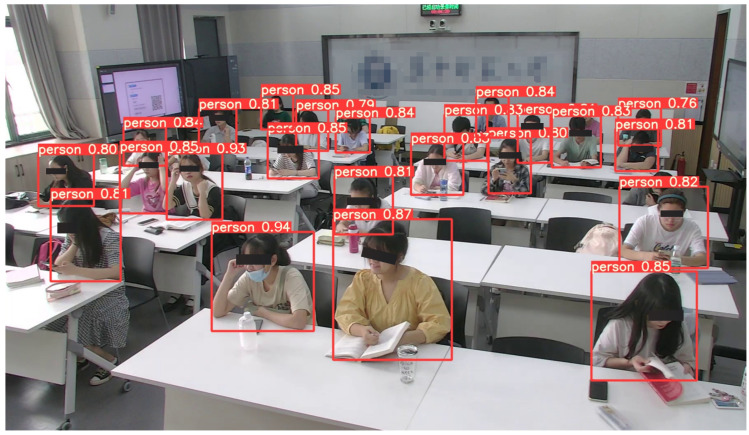
The results of student detection in the classroom.

**Figure 11 sensors-22-05932-f011:**
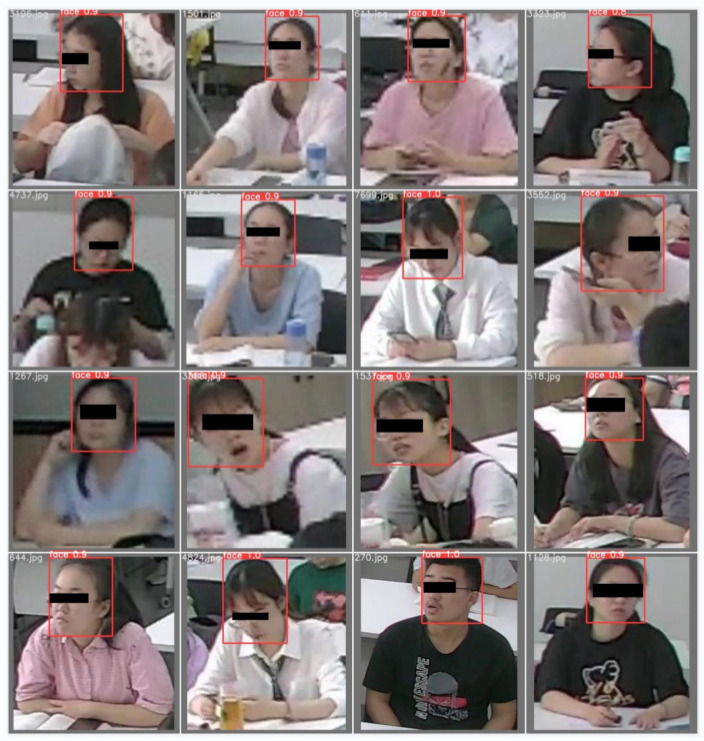
The results of face detection.

**Figure 12 sensors-22-05932-f012:**
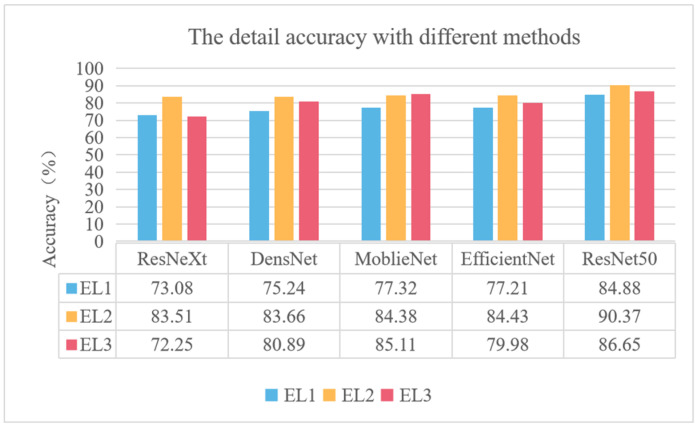
The detail testing accuracy with different methods.

**Figure 13 sensors-22-05932-f013:**
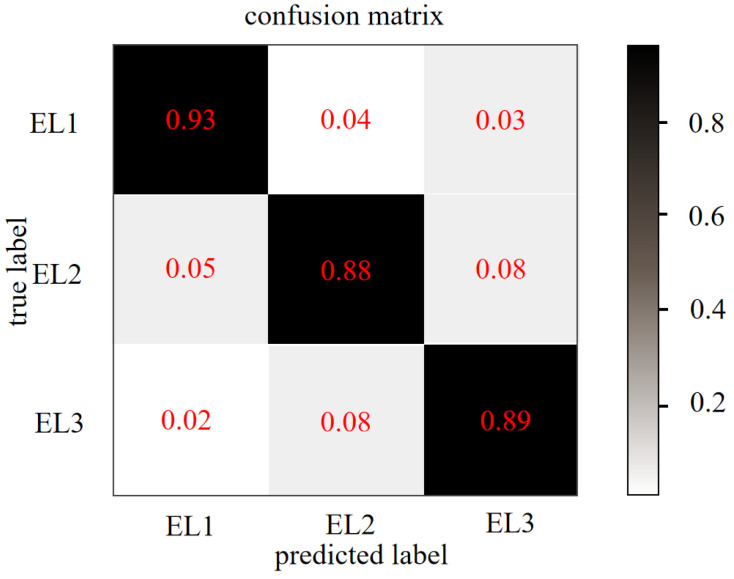
The confusion matrix based on the CoAtNet network.

**Table 1 sensors-22-05932-t001:** The comparison of YOLO algorithms in different versions.

YOLO Version	Backbone Feature Extractor	Loss Function	Neck
YOLOv1	GoogLeNet without inception	Sum-squared error	-
YOLOv2	Darknet-19	Sum-squared error	-
YOLOv3	Darknet-53	Binary cross entropy	FPN
YOLOv4	CSPDarknet-53	Binary cross entropy	SPP and PANet
YOLOv5	CSPDarknet-53	Binary cross entropy and logits loss function	FPN and PANet

**Table 2 sensors-22-05932-t002:** The distribution of the annotated samples.

Engagement Dimension	EL1	EL2	EL3
Emotional engagement	12,330	10,750	10,250
Behavioral engagement	8450	12,300	12,580
Overall engagement	12,850	12,100	8380

**Table 3 sensors-22-05932-t003:** The detection results of YOLOv5 algorithm with different loss functions.

Loss Function	Precision	Recall	AP	mAP@0.5	mAP@0.5:0.95
$L_{\mathrm{IoU}}$	89.8%	94.1%	97.4%	96.7%	65%
$L_{\mathrm{GIoU}}$	94%	88.6%	97.4%	96.9%	65%
$L_{\mathrm{CIoU}}$	87.6%	86.6%	95.4%	94.4%	64.6%
$L_{\mathrm{DIoU}}$	91.1%	94.8%	97.8%	97.8%	64.7%
$L_{\alpha -\mathrm{IoU}}$	95.4%	96.6%	98.4%	98.7%	65.4%
Improved Accuracy	1.4%	1.8%	0.6%	0.9%	0.4%

**Table 4 sensors-22-05932-t004:** The testing accuracy with different networks.

Learning Rate	ResNeXt	DensNet	MoblieNet	EfficientNet	ResNet50
**Accuracy**	76.28	79.93	82.27	80.54	**87.30**

**Table 5 sensors-22-05932-t005:** The accuracy of different networks trained on the self-built dataset.

The Method	VGG16 (Pretrained)	VGG16 (No-Pretrained)	ResNet18 (Pretrained)	ResNet18 (No-Pretrained)	CoAtNet
Accuracy	86.31	66.2	84.52	64.7	**89.97**

**Table 6 sensors-22-05932-t006:** This classification accuracy of different classifiers.

Classifier	DT	NB	KNN	LR	RF	SVM			
						Linear	Poly	RBF	Sigmoid
**Accuracy**	90.32	85.16	**90.91**	76.67	90.10	84.85	81.48	84.85	81.48

**Table 7 sensors-22-05932-t007:** The results were obtained using the different k.

The Value of k	Accuracy	Precision	Recall
k = 1	90.91	92.3	85.71
k = 2	93.94	92.86	92.86
k = 3	90.32	92.3	85.71
k = 4	87.88	85.71	85.71

## Data Availability

Not applicable.

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
