# Peer review of "Bimodal Learning Engagement Recognition from Videos in the Classroom"

_sensors, 2022, doi:10.3390/s22165932_

Round 1

Reviewer 1 Report

The article is within the scope of the journal. The topic described is interesting.

It is well written and structured, and easy to read.

The content of the article is original and shows progress in the area of knowledge.

The work methodology is exposed and an analysis of the results is carried out.

However there are some improvements to be accepted:

a) It would be necessary to extend the state of the art

b) It would be necessary to extend the discussion section in which the presented work is compared with other similar works and the limitations and advances of the work are indicated.

c) Some lines of future work should be indicated.

Reviewer 2 Report

-The author should explain more information about research hotspot in education.

-The author should explain about power IOU loss function in YOLOv5. and what is the full meaning of IOU ?

-The author should provide more detail about the architecture of the YOLOv5 model versus another YOLO model? and the author should bring compare table of the YOLO in this paper.

-How the residual model of ResNet does it work? and bring more detail. what are difference residual module and residual networks in deep neural networks?

-The author should explain about architecture of the CoAtNet Network can merges the convolution and self-attention within one basic computational block. 

and what is ConvNets and superior model that can be used in ConvNets? 

-The author should explain about Image samples with different engagement levels in the overall dimension. for example the student pretend to study How do we know that student is engagement in this class?

-The author should explain about students detection based on YOLOv5 in class. and how the grid at the center of student predict bounding box, confidence? 

For example if they put the something without students, How can it work?

-The author should check the equation again. So it looks like some mess up.

-The author should explain bimodal learning engagement recognition methodology work. and how emotional Engagement Net and Behavioral Engagement net work in

Learning engagement recognition?  

-How the mAP curve can increase sharply and tends to be stable? So the author should explain mAP under the experiments with different loss functions?

-The author should proof the reason of the test accuracy with different networks. and the author should proof the ResNet50 why is the highest for network with accuracy of 0.873.

Round 2

Reviewer 1 Report

The paper can be accepted in current form

Author Response

Thanks for your kind help and positive comments.

Reviewer 2 Report

The paper is well revised and can be accepted.

Author Response

(The authors gave the same response as above.)
